# Early Low-Fluence Red Light or Darkness Modulates the Shoot Regeneration Capacity of Excised *Arabidopsis* Roots

**DOI:** 10.3390/plants9101378

**Published:** 2020-10-16

**Authors:** Xi Wei, Yanpeng Ding, Ye Wang, Fuguang Li, Xiaoyang Ge

**Affiliations:** 1Henan Normal University Research Base of State Key Laboratory of Cotton Biology, Xinxiang 453000, China; chinaweixi521@163.com; 2State Key Laboratory of Cotton Biology, Institute of Cotton Research of CAAS, Anyang 455000, China; 15369229109@163.com (Y.D.); wangye01@caas.cn (Y.W.)

**Keywords:** low-fluence, red light, shoot regeneration, *WUS*, NPA

## Abstract

In plants, light is an important environmental signal that induces meristem development and interacts with endogenous signals, including hormones. We found that treatment with 24 h of low-fluence red light (24 h R) or 24 h of darkness (24 h D) following root excision greatly increased the frequency of shoot generation, while continuous low-fluence red light in callus and shoot induction stages blocked the explants’ ability to generate shoots. Shoot generation ability was closely associated with *WUS* expression and distribution pattern. 1-N-naphthylphtalamic acid (NPA) disrupted the dynamic distribution of the *WUS* signal induced by early 24 h R treatment, and NPA plus 24 R treatment increased the average shoot number compared with early 24 h R alone. Transcriptome analysis revealed that differentially expressed genes involved in meristem development and hormone signal pathways were significantly enriched during 24 R or 24 D induced shoot regeneration, where early 24 h R or 24 h D treatment upregulated expression of *WOX5*, *LBD*16, *LBD*18 and *PLT*3 to promote callus initiation and formation of root primordia, and also activated *WUS*, *STM*, *CUC*1 and *CUC*2 expression, leading to initiation of the shoot apical meristem (SAM). This finding demonstrates that early exposure of explants to transient low-fluence red light or darkness modulates the expression of marker genes related with callus development and shoot regeneration, and dynamic distribution of *WUS*, leading to an increased ability to generate shoots.

## 1. Introduction

Plant cells are pluripotent, meaning they have the potential to develop into an entire plant body from highly differentiated tissues or organs, or from a single somatic cell [1]. Explants have the ability to regenerate new root apical meristems (RAM) or shoot apical meristems (SAM) in the absence of sexual fertilization [2,3]. Regeneration in differentiation processes can be divided into two categories, including somatic embryogenesis and somatic organogenesis. Somatic organogenesis is important for transgenic plant generation [4,5]; shoot regeneration can be induced from callus tissues culture in two phases. In the first phase, explants of excised *Arabidopsis* root or cotyledon are cultured on a callus-induction medium (CIM) under dark conditions to induce callus formation [5]. Callus cells form when the plant tissue becomes dedifferentiated and acquires pluripotency, which is necessary for shoot regeneration [6,7]. Some studies have shown that callus initiation on a CIM is similar to the rooting pathway in non-root organs where the newly formed callus resembles a group of root primordium-like cells [8,9]. Auxin and cytokinin, involved in somatic organogenesis, may exhibit similar function to that in lateral root development, wherein auxin triggers lateral root initiation but cytokinin inhibits lateral root formation [10]. Furthermore, acquisition of pluripotency in callus cells is also regulated by PLETHORA3 (*PLT*3), *PLT*5, and *PLT*7 genes [11]. In the second stage, the callus is transferred to a shoot-induction medium (SIM) to induce the shoots. The shoot induction process consists of several critical events including the distribution of phytohormones over a gradient, initiation of the shoot meristem, and organ formation [1,12]. The CUP-SHAPED COTYLEDON 1 (*CUC*1) and *CUC*2, as No apical meristem, Arabidopsis transcription activation factor, Cup-shaped cotyledon(NAC) transcription factors, regulate the initiation of shoot meristem tissue and promote adventitious shoot regeneration by activating expression of *STM* [13]. A *cuc*1 *cuc*2 double knockout mutation impairs the capacity for shoot regeneration in the callus, while overexpression of *CUC*1 or *CUC*2 improves the capacity for shoot regeneration [11]. A specific ratio of auxin and cytokinin is key for ensuring *WUS* induction at an appropriate expression level during de novo shoot regeneration in *Arabidopsis* [1,12,14]. *WUS* expression is activated by cytokinin response regulatory factors B-type *ARABIDOPSIS* RESPONSE REGULATORs (*ARRs*) in regions with high levels of cytokinin, which leads to a cell fate transition from callus pluripotency cells to stem cells [15,16].

Light, a critical environmental signal, also modulates shoot regeneration, and has profound developmental effects on shoot organogenesis [17]. After tissue excision, low or high intensity light treatment can affect shoot regeneration in multiple plant species [18]. *Arabidopsis* explants are typically placed in continuous darkness or white light immediately after excision [6]. As for the effects of treatment with specific colors of light, some studies revealed that shoot regeneration was inhibited by blue/UV-A wavelengths, since high-energy wavelengths are absorbed by chlorophyll, thus leading to photosystem II damage [19]. Blue/UV-A wavelengths, even in low fluence light, can inhibit long-term shoot regeneration via a *CRY*1 photoreceptor-mediated signaling pathway [20]. High intensity light reduces the ability for shoot regeneration in *Arabidopsis* explants in vitro. Previous studies found that light affects multiple signaling pathways involving auxin [21], cytokinin [22], ethylene [23], red/far-red light photoactivation [24,25], blue/UV-A light photoactivation [26], and photo-oxidative damage [18]. Shoot regeneration was inhibited by treatment with 24 h blue/UV-A wavelengths after organ excision, while far red light signaling counteracts the inhibitory effects on shoot regeneration of early high intensity light exposure [20]. However, it is still not clear what mechanisms underly light regulation of adventitious shoot meristem formation as well as the role of early red light signaling on modulating the efficiency of shoot regeneration.

Red light, a component of sunlight, is of great importance for plant development. Exposure to red light significantly effects morphology, enzymatic activities, and the accumulation of bioactive compounds in *Anoectochilus roxburghii* [27]. The appropriate combination of red and blue wavelengths during embryogenic callus differentiation promotes somatic embryo maturation and conversion in sugarcane [28]. Low flux red light enhances the synthesis of endogenous auxin in *Arabidopsis* meristems [29,30]. We inferred from these reports that low flux red light may also play an important role during shoot organogenesis. In this study, we found that exposure of explants to long-term low-fluence red light strongly inhibited the generation of adventitious shoots, while 24 h exposure to low-fluence red light after root excision significantly improved the efficiency of shoot regeneration.

## 2. Results

### 2.1. Effects of Different Light Combinations on Shoot Regeneration Capacity

Root explants from wild-type Arabidopsis Col-0 were used to evaluate the effects of different combinations of light on the capacity for shoot regeneration, through treatments applied during the CIM and SIM stages (Figure 1A). The shoot regeneration capacity was calculated at 10 days, 14 days and 16 days on SIM following the different treatments, respectively. The conditions of the control culture for root explants were first callus induction under seven days of darkness on a CIM, followed by shoot generation on a SIM under white light (D-W). Compared with the control D-W, continuous red light treatment for 7 days on a CIM and white light treatment on a SIM (R-W) caused no significant differences in shoot formation at 10 days and 14 days of shoot induction, but showed an obviously decreased capacity for shoot generation at 16 days, where the percent of explants with shoots was substantially decreased compared to the control (Figure 1B,C).

To investigate the effects of red light on shoot regeneration during the SIM stage, callus was subjected to dark culture on CIM for 7 days followed by red light treatment in the SIM stage for 16 days (D-R). This treatment resulted in severe inhibition of shoot regeneration at 14 days and 16 days, suggesting that long-term red light treatment in the SIM stage was detrimental to shoot formation. As expected, continuous red-light treatment in CIM and SIM stages (R-R) blocked the capacity for shoot regeneration (Figure 1B,C), demonstrating that long-term exposure to red light weakens the ability to form shoots.

While long-term red light treatment produced negative effects on shoot regeneration regardless of the CIM or SIM stage, short-term light treatment showed contrasting effects. In this study, the light regimens of either 24 h dark (24 h D-W) or 24 h red light (24 h R-W) in the initial stage after root excision and then shifting to white light in the CIM and SIM was used to regulate the shoot generation. These treatments both induced shoot formation, whereas roughly 7.5% and 21% of explants regenerated shoots at 10 days in SIM following 24 h D-W or 24 h R-W, respectively (Figure 1C). No shoot formation was observed at 10 days in the SIM stage if either 7 days-culture in darkness or 7 days of red light treatment was used in the CIM stage (Figure 1C), indicating that the initial 24 h of treatment under dark or red light was sufficient for shoot induction. Subjection to the 24 h R-W treatment after root excision significantly increased the percentage of explants with shoots, where about 90% and 94.8% of explants regenerated shoots at 14 days and 16 days. Similarly, 24 h D-W treatment after root excision also promoted shoots regeneration, where about 41.8% and 50.3% of explants regenerated shoots at 14 days and 16 days (Figure 1C). In contrast, long term exposure to darkness or red light in the CIM stage decreased the potential for shoot generation, where about 20% and less than 40% of explants regenerated shoots at 14 days and 16 days (Figure 1C).

### 2.2. Effects of Different Light Regimens on Shoot Growth Vigor, Distribution Pattern and Shoot Regeneration Number per Explant

Based on the above results, darkness or red light during the initial 24 h after root excision significantly increased the percentage of explants with shoots, while the effect on the shoot number per explant, shoot growth vigor and shoot distribution pattern was not studied. Here, 120 individual explants in each replicate were collected, and the average shoot number per explant was calculated. As shown in Figure 2A,C, the shoot morphology and distribution pattern were distinctly different between these treatments. Specifically, red light treatment in the CIM and white light treatment in the SIM (R-W) significantly inhibited shoot growth, resulting in the development of small and abnormal shoots (Figure 2B). In contrast, regenerated shoot number and size increased following the control D-W, supporting that continuous darkness not continuous red light treatment in CIM facilitated the shoot development. The 24 h R-W and 24 h D-W treatment conditions both significantly increased the shoot generation frequency, how about the effect on shoot number per explant? Compared with the control D-W treatment, the 24 h R-W and 24 h D-W treatments presented the different effects on shoot number per explant, where the average shoot number per explant was slightly increased under 24 h D-W but decreased under 24 h R-W. Notably, 24 h R-W greatly improved shoot growth vigor and changed the shoot distribution patterns. Unlike the weak growth vigor and the wide distribution pattern of shoots under D-W, R-W and 24 h D-W conditions, most shoots generated after 24 h R-W treatment developed into seedling with multiple normal leaves, and emerged from the middle location of the callus after the 24 h R-W treatment (Figure 2A,C).

### 2.3. NPA Treatment Disrupts the Red Light Induced Shoot Distribution Pattern and Changes the Shoot Number per Explant

Shoot generation was closely associated with the *WUS* expression location and auxin distribution, *WUS* is expressed in the region of low auxin level, and high auxin levels were around the area of *WUS* expression [31]. The guiding hypothesis of this work is that 24 h R-W may regulate the pattern of shoot distribution by controlling the *WUS* location depending on the polarity of auxin distribution. To test this hypothesis, auxin transport inhibitor 1-N-naphthylphtalamic acid (NPA) was added to the CIM at three different concentrations during the early treatment of 24 h red light. Root explants were subsequently transferred to the CIM without NPA for callus induction under white light. Compared with the 24 h red light treatment lacking NPA, the addition of NPA substantially changed shoot numbers and patterns of shoot distribution. Shoot numbers at 16 d on the SIM averaged 3.4 shoots for each explant under 24 h R-W treatment, while the average shoot number per explant increased to 4.4, 5.6, and 7.8 shoots per explant with the addition of 12.5 μM NPA, 25 μM NPA, 50 μM NPA, respectively (Figure 3C). The shoot distribution patterns were also disrupted, the generated shoots were centralized to the middle location of callus under the 24 h R-W treatment, while the shoots were widely distributed around the callus after NPA treatment (Figure 3A,B). A likely cause of this altered phenotype is that NPA interferes with the transport and distribution of auxin that is otherwise regulated by 24 h R-W treatment, the widely spread auxin gradients may facilitate the distribution of the *WUS* signal, thus leading to wider patterns of shoot distribution and an increased average number of shoots.

### 2.4. Dynamic Distribution of WUS under Different Light Regimens and NPA Treatment

The capacity for shoot regeneration is controlled by the level of *WUS* expression, while the location of *WUS* expression determines where and when shoots merge from the callus [14,32]. For the purpose of detecting the location of WUS expression, and thus shoot distribution under different treatments, a pWUS::*WUS*-GUS marker line was used. As shown in Figure 2, the average shoot number per explant after 24 h D-W treatment slightly increased over that of the standard D-W treatment, which is a phenomenon closely associated with *WUS* signal strength and distribution patterns. Specifically, *WUS* signal strength and distribution area across callus cells after the 24 h D-W treatment was slightly increased in comparison with callus cells subjected to the D-W treatment (Figure 4A,B). Expectedly, the weak *WUS* signal under R-W treatment caused a reduction in shoot number and inhibition of shoot growth, whereas the centralized strong *WUS* signal observed under the 24 h R-W treatment promoted shoot growth vigor and the centralized distribution of shoots (Figure 4A,B).

Given that NPA treatment promotes an increase in shoot number per explant following 24 h red light conditions, we further hypothesized that the *WUS* distribution pattern was changed. Similar to the shoot distribution patterns, *WUS* expression was observed to be centralized to specific locations following 24 h red light treatment. NPA treatments significantly disrupted the distribution of *WUS*, and WUS signals were widely expressed in the callus after 7 days on the SIM (Figure 4A,B). Moreover, the WUS signals and distribution area gradually increased commensurately with increased NPA concentration (Figure 4A,B). These results support that WUS expression patterns thus appeared to be regulated by red light, darkness, duration of light treatment, and auxin polar distribution.

### 2.5. Expression of Marker Genes Involved in Shoot Regeneration and Callus Development Are Dynamically Regulated by Light and NPA

Stem cells within the SAM are necessary during organogenesis and somatic embryogenesis, and in these cells *WUS* gene expression is critical for the regulation of stem cell fate [32]. The pre-incubation stage on the CIM was necessary to activate *WUS* expression to regulate stem cell fate in the SIM stage as described by Shemer et al. [33]. Different to the low expression level of WUS in the CIM, *WUS* expression was significantly induced at 7 days on SIM after the above five treatments, whereas 24 h R-W and 24 h D-W treatments both significantly activated *WUS* expression compared with other treatments (Figure 5A). This finding suggests that high levels of *WUS* expression promoted SAM initiation and shoot formation via regulation of stem cell fate, shown by an increased shoot generation frequency at 10 days, 14 days and 16 days on the SIM after 24 h R-W and 24 h D-W treatments (Figure 1C).

SHOOT MERISTEMLESS (STM) and organ boundary genes CUP SHAPED COTYLEDON1 (*CUC*1), *CUC*2, and *CUC*3 regulate each other to establish the embryonic SAM and to specify cotyledon boundaries during embryogenesis [13,34]. Compared with the D-W, R-W and 24 h D-W treatments, the 24 h R-W treatment obviously upregulated *STM*, *CUC*1 and *CUC*2 expression (Figure 5A), which supports the data showing that the 24 h R-W treatment, after root excision significantly increased the percentage of shoot-bearing explants (up to 94.8%) at 16 days on the SIM (Figure 1B,C). We concluded from these data that shoot formation ability was regulated by the expression level of marker genes depending on which light treatment was applied. Specifically, the higher expression level of *WUS*, *STM*, *CUC*1 and *CUC*2 under the 24 h R-W treatment compared to other treatments significantly promoted the shoot generation capacity.

*WOX*5, *PLT*3, *LBD*16 and *LBD*18 are also key genes controlling callus development. The 24 h R-W and 24 h D-W treatments significantly upregulated *WOX*5 expression relative to the other five treatments at CIM7 (Figure 5B), suggesting that high levels of *WOX*5 expression provided the basis for induction of *WUS* expression, coinciding with the high shoot regeneration rates under the 24 h R-W and 24 h D-W treatments (Figure 1). Compared to CIM 0, *PLT*3, *LBD*16 and *LBD*18 both presented high expression levels at CIM7 under all seven different treatments (Figure 5B), suggesting that *PLT*3, *LBD*16 and *LBD*18 play key roles in mediating the formation of root primordia, which thus provides the foundation for stem cell formation. Primer sequences of marker genes for callus-induction and shoot-induction used in the Appendix A during this study.

### 2.6. DEGs in CIM and SIM Stages under D-W, 24 D-W and 24 R-W Treatments

Early low-fluence red light or darkness facilitates the shoot regeneration of excised Arabidopsis roots (Figure 1B,C), the samples at CIM0, CIM7, SIM7 under D-W, 24 D-W and 24 R-W treatments were selected to reveal the regulatory mechanism through transcriptome analysis. Analysis of Pearson’s correlation coefficient confirmed that the high repeatability among the three biological samples of CIM0, DWCIM7, DCIM7, RCIM7, DWSIM7, DSIM7 and RSIM7 (Appendix A). The three period materials were obviously clustered into three groups including group one (CIM0), group two (DWCIM7, DCIM7, RCIM7) and group three (DWSIM7, DSIM7, RSIM7) (Appendix A). The Venn diagram reflected that 14,482 genes were co-expressed in the CIM stage, but 74, 92 and 195 genes were exclusively expressed in RCIM7, DCIM7 and DWCIM7 (Appendix A), respectively. A total of 15,790 genes were co-expressed in the SIM stage, 226, 259 and 135 genes were exclusively expressed in RSIM7, DSIM7 and DWSIM7, respectively (Appendix A). These results suggest that the shoot regeneration ability was controlled by a large number of common genes and a few private genes.

Differentially expressed genes (DEGs) between different stages and treatments are listed in Appendix A, and DEGs induced by light signal was listed in Appendix A. We found that auxin-responsive genes, *IAAs* and *ARFs* were expressed in the CIM stage not in the SIM stage (Appendix A). Consistent with the results of quantitatively detected marker genes (Figure 5A,B), *PLTs*, *WOX*5, *WOX*11, *LBD*16, *LBD*18, *LBD*19 related with root primordia properties were activated during the CIM stage, but restricted expression in the SIM stage (Appendix A). Besides, expression of auxin efflux carrier *PIN*1, *PIN*7 and embryogenesis related genes *BBM*, *AGL*15 were obviously induced in the CIM and SIM stages (Appendix A). However, *LEC*1, *LEC*2, *ABI*3, *FUS*3 related to embryogenesis were still restricted in the CIM stage and in the primary regeneration shoot stage (Appendix A).

### 2.7. GO and KEGG Enrichment Analysis of DEGs in the CIM and SIM Stage

In order to study the function of DEGs, gene ontology (GO) and the Kyoto Encyclopedia of Genes and Genomes (KEGG) enrichment analysis were performed. Some DEGs were involved in the GO terms, including meristem development, embryo development; plant hormone response, transport, biosynthesis, signal; cellular response to red light, far red light, dark (Appendix A). For the DEGs between DWCIM7, DWSIM7, RCIM7, RSIM7, DCIM7 and DSIM7, the GO terms “cell differentiation”, ”maintenance of shoot apical meristem identity”, “stem cell population maintenance” and ”shoot apical meristem specification” were significantly enriched (Appendix A). For the DEGs between DWCIM7, RCIM7, DCIM7, the GO terms: “response to red light or far red light”, ”cellular response to light stimulus” were significantly enriched (Appendix A). For the DEGs between DWCIM7 and CIM0, GO terms related with plant hormone signaling, including: “response to auxin”, ”auxin polar transport”, ”abscisic acid transport” and “response to cytokinin, jasmonic acid and oxygen signal” were significantly enriched (Appendix A).

KEGG pathway enrichment analysis of DEGs showed that the plant hormone signal pathway, starch and sucrose metabolism and fatty acid elongation were significantly enriched among different treatments. For the transitional process from callus to shoot regeneration, 3637 common DEGs, 1111 and 1085 private DEGs were detected between DCIM7 vs. DSIM7 and RCIM7 vs. RSIM7 (Appendix A), and these DEGs involved in the KEGG pathways: “Plant hormone signal transduction”, ”Brassinosteroid biosynthesis”, ”Fatty acid elongation”, ”Starch and sucrose metabolism” were significantly enriched (Appendix A). For the CIM7 stage (callus dedifferentiation), 22 DEGs between DCIM7 and RCIM7, 607 DEGs between DWCIM7 and DCIM7, 424 DEGs between DWCIM7 and RCIM7 were detected (Appendix A), and these DEGs involved in KEGG pathways, “Phenylpropanoid biosynthesis” and ”Indole alkaloid biosynthesis” were significantly enriched (Appendix A). For the SIM7 stage (regeneration shoot), 541 DEGs between DSIM7 and RSIM7, 867 DEGs between DWSIM7 and DSIM7, 266 DEGs between DWSIM7 and RSIM7 were detected (Appendix A), and these DEGs involved in KEGG pathways: “Phenylpropanoid biosynthesis”, ”Protein processing in endoplasmic reticulum”, ”alpha-Linolenic acid metabolism”, and ”Starch and sucrose metabolism” were significantly enriched (Appendix A). Some representative DEGs involved in enriched KEGG pathways are summarized in Table 1 (transitional stage, DCIM7 vs. DSIM7, RCIM7 vs. RSIM7), Table 2 (dedifferentiation stage, DWCIM7 vs. DCIM7, DCIM7 vs. RCIM7, DWCIM7 vs. RCIM7), Table 3 (regeneration shoot, DWSIM7 vs. DSIM7, DWSIM7 vs. RSIM7, DSIM7 vs. RSIM7).

Based on the GO and KEGG analysis results, we proposed a possible model for revealing the mechanism controlling the capacity of shoot regeneration and callus formation under the early low-fluence red light or darkness (Figure 6). Appendix A was the overall situation of KEGG enrichment pathway genes corrected in the transition stage (DCIM7 vs. DSIM7, RCIM7 vs. RSIM7). Appendix A was the overall situation of KEGG enrichment pathway genes corrected in the dedifferentiation stage (DWCIM7 vs. DCIM7, DCIM7 vs. RCIM7, DWCIM7 vs. RCIM7). Appendix A was the overall situation of KEGG enrichment pathway genes corrected in the primary regeneration shoot stage (DWSIM7 vs. DSIM7, DWSIM7 vs. RSIM7, DSIM7 vs. RSIM7). In the callus induction (Figure 6A) and shoot regeneration process (Figure 6B), plant hormones transduction including auxin signaling, jasmonate signal transduction, brassinosteroid signal transduction, gibberellic acid mediated signaling, abscisic acid(ABA)-induced signal transduction and cytokinin signal transduction were enriched (Appendix A). Meanwhile, we found that fatty acid elongation, brassinosteroid biosynthesis, red light signaling, starch and sucrose metabolism, carbon fixation, carbon metabolism and cutin, suberine and wax biosynthesis also participated in the process of shoot regeneration.

## 3. Discussion

### 3.1. Early Red Light or Dark Exposure on Excised Root Tissue Improved the Shoot Regeneration Capacity via Regulation of WUS Signal Strength and Distribution Pattern

Explants, historically, are often placed in a continuous dark or light treatment immediately after excision in *Arabidopsis* [6]. In terms of monochromatic light treatments, some studies revealed that shoot regeneration was inhibited by blue/UV-A wavelengths [19]. Additionally, low far red (FR)reduced chloroplast xanthophyll pigments, and was not sufficient to elicit ROS, leading to inhibition of shoot regeneration [35].

In this study, our data suggest that root explants may be highly susceptible to conditions in the initial 24 h following excision. We found a significant benefit was gained from exposure to darkness or low-fluence red light treatment during the first 24 h after root excision, leading to an increase in the shoot regeneration frequency over that of callus exposed to continuous darkness in the CIM stage (Figure 1B,C). These results support data showing that darkness or low-fluence red light exposure during the initial 24 h after root excision regulates long-term shoot regeneration in *Arabidopsis* ecotype Columbia. Low-fluence red light increased the biosynthesis and transport of free indole-3-acetic acid (IAA) in the *Arabidopsis* meristem, cotyledons, hook and hypocotyl, and also promoted auxin biosynthesis in cucumber seedlings [29,30,36]. We found that the accumulation of auxin in the dark treatment for 3 days (D-3d) was lower than that after 24 h of red light treatment and continued white light treatment until 3 days (R-3d) (Appendix A). Similar results were observed at 5 days between D-5d and R-5d (Appendix A). Low-fluence red light may promote shoot regeneration because of the accumulation of auxin in the early stage. The capacity for shoot generation reported to be closely related with the endogenous auxin gradient [1]. So, we hypothesized that the initial treatment with 24 h of low-fluence red light may change the auxin distribution gradient via regulating auxin biosynthesis and transport, and finally promote shoot generation frequency. To test this hypothesis, NPA treatments were used to block the polar transport of endogenous auxin, average shoot number per explant was increased along with the changed auxin gradient. A likely cause of this altered shoot number was that altered auxin polar gradient regulated the *WUS* polar distribution.

The induction of *WUS* is the most critical event in the shoot formation phase, which is controlled by interaction of auxin and cytokinin [37]. Auxin-induced *WUS* expression is essential for embryonic stem cell renewal during somatic embryogenesis and de novo shoot regeneration in *Arabidopsis* [12,31]. Here we show that *WUS* distribution pattern and signal strength were both changed by the initial 24 h red light or 24 h darkness treatment in comparison with the control D-W treatment. In this case, *WUS* was localized in the middle location of the callus, which lead to a centralized distribution of shoots and increased shoot growth vigor. Different from the 24 h red light treatment, the 24 h darkness treatment increased the *WUS* signal distribution area, which increased the average shoot number per explant and changed the shoot distribution pattern. We believe that early 24 h red light or 24 h darkness treatments both modulated the polarity of auxin distribution but caused the different auxin distribution pattern. The auxin distribution pattern induced by the 24 h red light treatment led to centralized expression and localization of *WUS*.

### 3.2. Low-Fluence Red Light Increased the Capacity for Shoot Regeneration Depending on Upregulation of WOX5, LBD16, LBD18, PLT3, WUS, STM, CUC1, and CUC2

*LBD16* and *LBD18* may function redundantly in the establishment of a root primordium-like identity in the newly formed callus. Induction of *LBD16* on the CIM was found to be necessary to gain pluripotency in the callus, which thus modulated the ability for shoot generation, while inhibition of *LBD16* expression blocked the capacity for shoot development in the callus [8]. Higher expression of *LBD16* promoted the gain of callus pluripotency, resulting in formation of root founder cells. Subsequent activation of *WOX5* and *PLT3* synergistically, was previously reported to promote the fate transition from root founder cells to root primordium cells [11]. In this study, low-fluence red light upregulated *WOX5*, *LBD16*, *LBD18*, and *PLT3* expression, thus promoting the gain of callus pluripotency and root primordium formation.

The initial treatment condition of 24 h of red light after root excision significantly upregulated the expression levels of *WUS*, *STM*, *CUC*1, and *CUC*2, relative to their expression following the other treatments at 7 days on the SIM (Figure 5), suggesting that the simultaneous high expression of these four genes synergistically increased the capability for shoot regeneration. The *WUS* gene is critical for regulation of stem cell fate in plants, and low-fluence red light first induces high *WUS* expression during the SIM stage. This high expression of *WUS* specifies stem cell fate to promote the initiation of the SAM. Subsequently, expression of *CUC*1 and *CUC*2 are functionally redundant in the induction of SAM formation, through activation of the *STM*. Similar studies have also shown that overexpression of *CUC*1 and *CUC*2 genes in *Arabidopsis* promoted adventitious shoot formation on callus tissue via activation of *STM* expression [34], while *GhWUS* from *Gossypium hirsutum* promoted de novo shoot regeneration in *Arabidopsis* calluses by directly activating *CLV*3 and *CUC*2 [14]. Taken together, these data show that low-fluence red light promoted the expression of callus development genes *WOX*5, *PLT*3, *LBD*16, and *LBD*18, and also activated the shoot generation marker genes *WUS*, *STM*, *CUC*1, and *CUC*2, leading to a capacity for high shoot regeneration.

Our results indicate that the initial 24 h of treatment under dark or red light was sufficient for shoot induction. Treatment with NPA increased the average shoot number and caused wider distribution of shoots on calluses, a likely cause of this phenotype was that the dynamic distribution pattern of *WUS* expression was disrupted by the endogenous auxin gradient. However, the regulatory mechanism of *WUS* expression and dynamic distribution by red light remain to be elucidated. Increasing evidence suggests that red light affects auxin synthesis and transport [29,30]. Thus, it is possible that red light regulates *WUS* expression level and location depending on the auxin polar distribution, and the regulatory network needs to be further established.

### 3.3. GO and KEGG Enrichment Analysis of DEGs Discover the Vital Regulatory Pathway Underlying Early Low-Fluence Red Light or Darkness

GO enrichment analysis found that DEGs are mainly involved in meristem development, cell differentiation, response to red light or far red light, response to auxin, and auxin polar transport. KEGG enrichment analysis showed that plant hormone signal transduction, carbon metabolism, starch and sucrose metabolism, fatty acid elongation, brassinosteroid biosynthesis pathways were significantly enriched. Appendix A was FPKM values of all genes. And the Appendix A: All databases for gene annotation. These significantly enriched pathways and GO terms mainly included auxin response and transport genes (*IAAs*, *PINs*, and *ARFs*), meristem development genes (*WOX*5, *PLT*3, *LBD*16, *WOX*11), fatty acid elongation genes (*KCSs*), brassinosteroid biosynthesis genes (*BAS1*), suggesting that these above DEGs regulate shoot generation capacity controlled by early low-fluence red light or darkness. The callus initiation and gain of pluripotency is regulated by a number of transcription factors such as *WOX*5, *WOX*11, *WOX*12 and *LBDs* [9,11,38,39]. Very-long-chain fatty acids (*KCS*1) restrict regeneration capacity by confining pericycle competence for callus formation in *Arabidopsis* [40].

We also predicted other types of transcription factors, ERF, AP2, RSK, ARF, BES1, BSD, BUB, IAA and so on (Appendix A), these DEGs provided potential genes for establishing the regulatory network of shoot development, and their differential expression may play an important role in *Arabidopsis* shoot regeneration. Our research was the tip of the iceberg regarding the results of transcriptome analysis. Therefore, the mechanism controlling the capacity of shoot regeneration and callus formation under the early low-fluence red light or darkness needs further study.

## 4. Materials and Methods

### 4.1. Plant Materials

The *Arabidopsis thaliana* plants used in this study were of the wild-type Columbia (Col-0) genetic background. The pDR5::GUS::GFP line, which reflects the auxin level by monitoring auxin responsiveness, was used for Western blot analyses. The pWUS::WUS-GUS marker line was kindly provided by Professor Lin Xu (Institute of Plant Physiology and Ecology, China Academy of Science, China).

### 4.2. Plant Growth and In Vitro Culture

The *A. thaliana* seeds were surface-sterilized in tubes and then spread on seed germination Murashige and Skoog (MS) medium (1× MS salts, 2% sucrose, 0.3% Gelrite gellan gum, pH 5.7). The plates were kept at 4 °C in darkness for 48 h to overcome seed dormancy, after which they were placed in a greenhouse at 20–22 °C under a 16 h light/8 h dark photoperiod for two weeks. To collect the excised roots as explants, seedlings at 14 d post-germination were cut and root tissue was subsequently cultured on solid callus induction medium (CIM; 1× Gamborg’s B5 salts, 3% sucrose, 0.5 g/L MES, 0.05 mg/L kinetin, 0.5 mg/L 2,4-D, and 0.3% Gelrite gellan gum, pH 5.7) under different light combinations for 7 days at a constant temperature of 22 °C. After cultivation for 7 days, the explants were transferred into the shoot induction medium (SIM; 1× MS salts, 1% sucrose, 0.5 mg/L MES, 2 mg/L zeatin, 1 mg/L d-biotin, 0.4 mg/L IBA, 0.3% Gelrite gellan gum, pH 5.7) under continuous light at 22 °C. The shoots on each explant were defined as being at least 1 mm long. Shoot regeneration frequency was obtained by measuring the rate of the number of explants with shoot derived from total number of explants cultured on SIM. Average shoot number per explant were calculated by measuring the rate of the total number of shoots derived from the number of explants with shoots. The shoot regeneration frequency under different light combination treatments was calculated at 10, 14 and 16 days and the shoot number per explant was calculated at 16 days after transfer to the SIM medium. The experiments were performed with three biological replicates, each containing 120 root segments of *Arabidopsis thaliana*.

### 4.3. Lighting Conditions

Twenty-four hours of darkness, continuous high white light (photosynthetic photon flux density (PPFD): 80–90 µmol m^−2^ s^−1^) and constant temperature were provided by a cold light source plant growth box, which is required for the induction of callus. An LED plant growth lamp was used to induce shoots and provided a suitable intensity of red light (PPFD: 40–60 µmol m^−2^ s^−1^). Different light combinations were used during the culture process (Figure 1A). Low-fluence red light treatment means continuous red light for 24 h photoperiod, darkness treatment means continuous dark for 24 h photoperiod, white light treatment means continuous white light for 24 h photoperiod. D-W (the control treatment), dark in CIM, white light in SIM. R-W, low-fluence red light in CIM and white light in SIM. D-R, dark in CIM and low-fluence red light in SIM. R-R, low-fluence red light in both the CIM and SIM. The 24 h D-W treatment involved early 24 h dark and then shifting to 6 days white light in the CIM followed by white light throughout the SIM. While the 24 h R-W treatment involved early 24 h low-fluence red light shifting to 6 days white light in the CIM, followed by white light treatment in the SIM. After different combinations of light treatment, samples at CIM 0 d, CIM 7 d (SIM 0 d), and SIM 7 d were collected for qRT-PCR analysis.

### 4.4. Western Blot Analyses

The pDR5::GUS::GFP transgenic plants were used to reveal GFP protein expression levels. GFP-fusion transgenic plants were used for Western blot analyses with anti-GFP antibodies. The ratio of gray values, which reflects the relative expression of protein, was equal to the ratio of the gray value of the GFP protein to the gray value of the internal control. The ACTIN protein was used as an internal control. The software ImageJ was used to calculate the gray value.

### 4.5. Total RNA Isolation and Quantitative Real-Time (qRT)-PCR Analysis

Total RNAs was isolated from the samples collected. The PrimeScriptTMRT reagent Kit with gDNA Eraser (TaKaRa) was used to remove genomic DNA from the total RNA and to obtain cDNA. The sequences of all the qRT-PCR primers are provided in Appendix A. The cDNA was diluted four to five times and then used as a template for the qRT-PCR. For the qRT-PCR, actin2 (AtACT2, AT3G18780) was used as an internal standard, and the gene expression level of CIM day 0 was set to 1 for quantification of relative expression. Three biological replicates were carried out for this experiment, and 30 individual calluses as a biological replicate were collected for qPCR expression analysis.

### 4.6. RNA-Seq Analysis

After different combinations of light treatment, samples at CIM0 (CIM 0 d), RCIM7 (24 h R-W treatment, CIM 7 d), DCIM7 (24 h D-W treatment, CIM 7 d), DWCIM7 (D-W treatment, CIM 7 d), RSIM7 (24 h R-W treatment, SIM 7 d), DSIM7 (24 h D-W treatment, SIM 7 d) and DWSIM7 (D-W treatment, SIM 7 d) were collected for RNA-seq. Genes with a log2 fold change ≥ 2 were classified as being significantly up-regulated/down-regulated in samples at CIM0, RCIM7, DCIM7, DWCIM7, RSIM7, DSIM7 and DWSIM7. RNA-Seq expression was standardized as fragments per kilobase million (FPKM).

### 4.7. Chemical Inhibitor

Different concentrations of 1-N-naphthylphtalamic acid (NPA) were used as follows: 12.5 μM, 25 μM, 50 μM 1-N-naphthylphtalamic acid (NPA) and dimethyl sulfoxide (DMSO). The NPA was dissolved in DMSO, which was filter-sterilized and added to CIM after autoclaving. The roots of 14-day-old seedlings (post germination) were excised and placed on CIM containing NPA for 24 h of culture under low-fluence red light. Seedlings were then transferred to CIM without NPA for 6 days under white light and finally transferred to SIM for shoot induction. The experiments were performed with three biological replicates, each containing 100 to 120 root segments of *Arabidopsis thaliana*.

### 4.8. β-GUS Assay

GUS chemical tissue staining experiments were performed according to protocols in a previous study [8]. To clearly observe the GUS staining, the stained tissues were decolorized with an alcohol concentration gradient. Under the stereo microscope, the gene expression level and localization of expression were observed in specific tissues via GUS staining. In this study, the pWUS::WUS-GUS marker lines were used to perform tissue staining after 7 days of induction on SIM. The above experiments were performed with three biological replicates, each containing 120 root segments, callus derived from root segment were stained.

## 5. Conclusions

The results of this work revealed that early low-fluence red light or darkness promotes the shoot regeneration capacity of excised *Arabidopsis* roots. NPA treatment disrupts the red light induced shoot distribution pattern and changes dynamic distribution of *WUS*. The 24 h D-W and 24 h R-W treatments obviously upregulated expression of marker genes involved in shoot regeneration and callus development, such as *WUS*, *STM*, *CUC*1, *WOX*5 and *LBD*16. GO and KEGG enrichment analysis found that DEGs are mainly involved in meristem development, cell differentiation, response to red light or far red light, response to auxin, auxin polar transport and plant hormone signal transduction, carbon metabolism, starch and sucrose metabolism, fatty acid elongation, brassinosteroid biosynthesis pathways. The findings of this study provided fundamental evidence into the mechanism of shoot regeneration, which will support future functional examination of vital molecular mechanisms of shoot regeneration.

## Figures and Tables

**Figure 1 plants-09-01378-f001:**
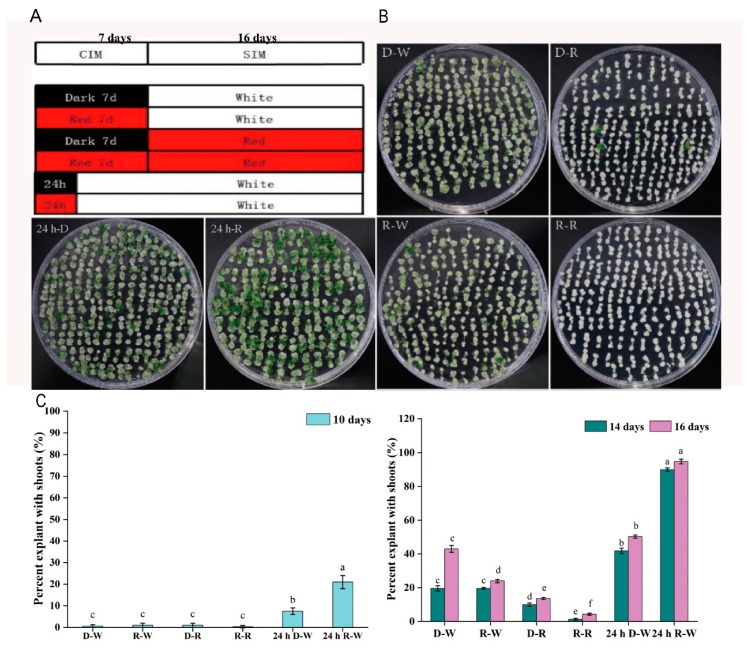
Effects of treatments with combinations of darkness, red, and white light on shoot regeneration in *Arabidopsis* Col-0 callus. (**A**) Light-combinations used in the callus-induction medium (CIM) and shoot-induction medium (SIM) stages. (**B**) Phenotypes of shoots induced under different light treatments at 16 days on the SIM. (**C**) Percent of explants with shoots at 10, 14, and 16 days on the SIM. D-W (the control treatment), dark in the CIM, white light in the SIM. R-W, red light in the CIM and white light in the SIM. D-R, dark in the CIM and red light in the SIM. R-R, red light in both the CIM and the SIM. 24 h D-W, early 24 h dark and then shifting to 6 days white light in the CIM followed by white light throughout the SIM. 24 h R-W, early 24 h red light shifting to 6 days white light in the CIM, followed by white light treatment in the SIM. The above experiments were performed with three biological replicates, each containing 120 root segments of *Arabidopsis thaliana*. Standard errors were calculated from three sets of biological replicates. A significant difference in the percent of explants with shoots between different treatments was analyzed at 10 days, 14 days and 16 days, respectively. The least significant difference method (LSD) was used for significance test (*p* < 0.05); different lowercase letters represent statistical differences in pairwise comparisons between LSD test groups (*p* < 0.05).

**Figure 2 plants-09-01378-f002:**
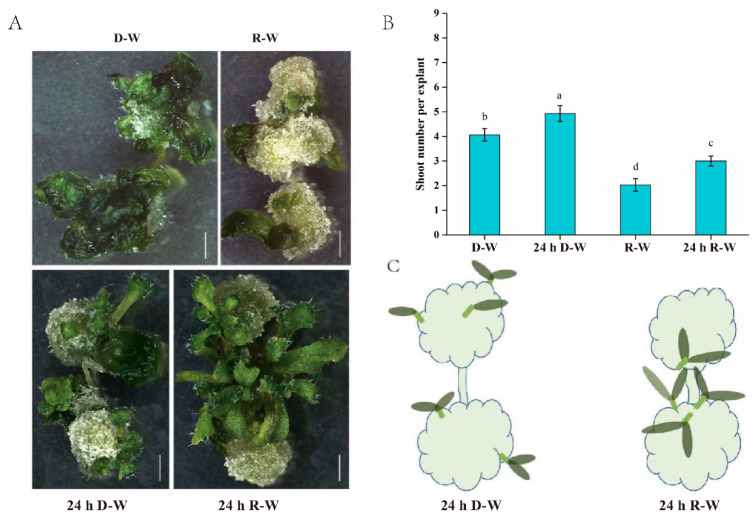
Effects of different light regimens on shoot regeneration number per explant, shoot growth vigor and shoot distribution. (**A**) Shoot morphology of explants under different treatments. The shoot growth vigor was severely inhibited under the R-W treatment, but was promoted under the 24 h R-W treatment. (**B**) Shoot number per explant under different treatments. In contrast with other treatments, average shoot number per explant under R-W treatment was significantly decreased. (**C**) Shoot distribution pattern under different treatments. The shoots were centralized to the middle location under the 24 h R-W treatment, but were widely distributed around callus under the 24 h D-W treatment. The shoot distribution pattern under the D-W or R-W treatments was similar to that of the 24 h D-W treatment. Error bars indicated the standard deviation from three independent experiments, each containing 120 root segments of *Arabidopsis thaliana*. The least significant difference method (LSD) was used for the significance test (*p* < 0.05); different lowercase letters represent statistical differences in pairwise comparisons between LSD test groups (*p* < 0.05).

**Figure 3 plants-09-01378-f003:**
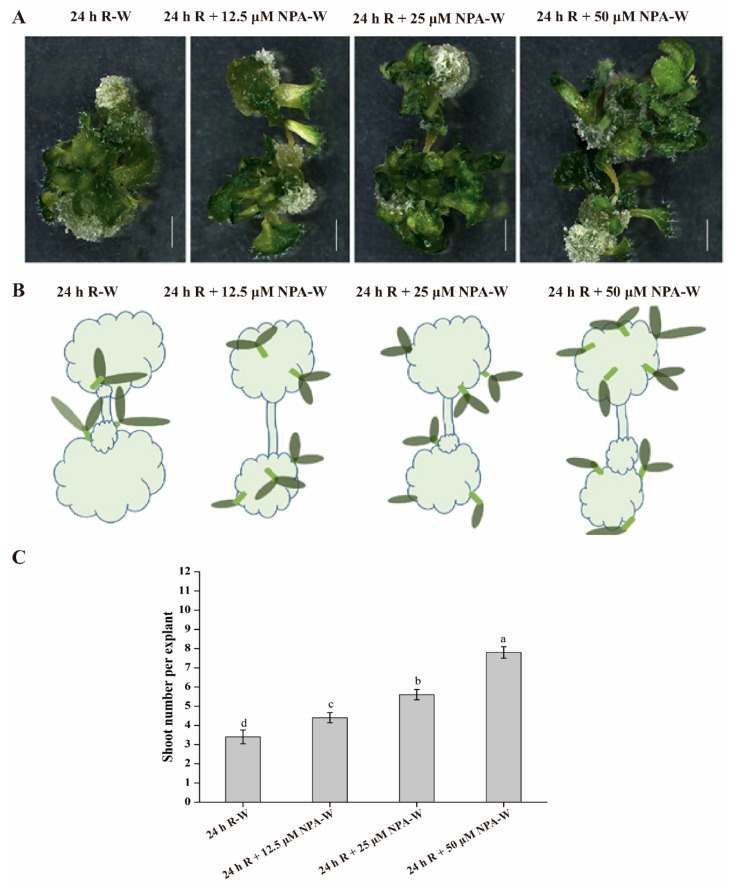
Effects of 1-N-naphthylphtalamic acid (NPA) treatment on average shoot number per explant, the percent explant with shoots and shoot distribution pattern. (**A**) Shoot number and distribution pattern under different treatments. Compared with the shoot morphology under the 24 h R-W treatment, different concentrations of NPA both increased the average shoot number and distributed the shoot distribution pattern. (**B**) Model for shoot distribution pattern under different treatments. Shoots were centralized to the middle location under the 24 h R-W treatment, NPA treatments caused the wide distribution of shoots. (**C**) Effects of NPA concentration on average shoot number per explant. With the increase in NPA concentration, the average shoot number per explant was also increased. The 24 h R-W treatment refers to early 24 h red light shifting to 6 days white light in the CIM, followed by white light treatment in the SIM. The 24 h R + 12.5 μM NPA-W, 24 h R + 25 μM NPA-W, or 24 h R + 50 μM NPA-W treatments refer to: 24 h of red light treatment on CIM containing 12.5, 25, or 50 μM NPA after root excision, then transfer to white light in the CIM and SIM. Error bars indicate the standard deviation from three independent experiments, each containing 120 root segments of *Arabidopsis thaliana*. The least significant difference method (LSD) was used for the significance test (*p* < 0.05); Different lowercase letters represent statistical differences in pairwise comparisons between LSD test groups (*p* < 0.05).

**Figure 4 plants-09-01378-f004:**
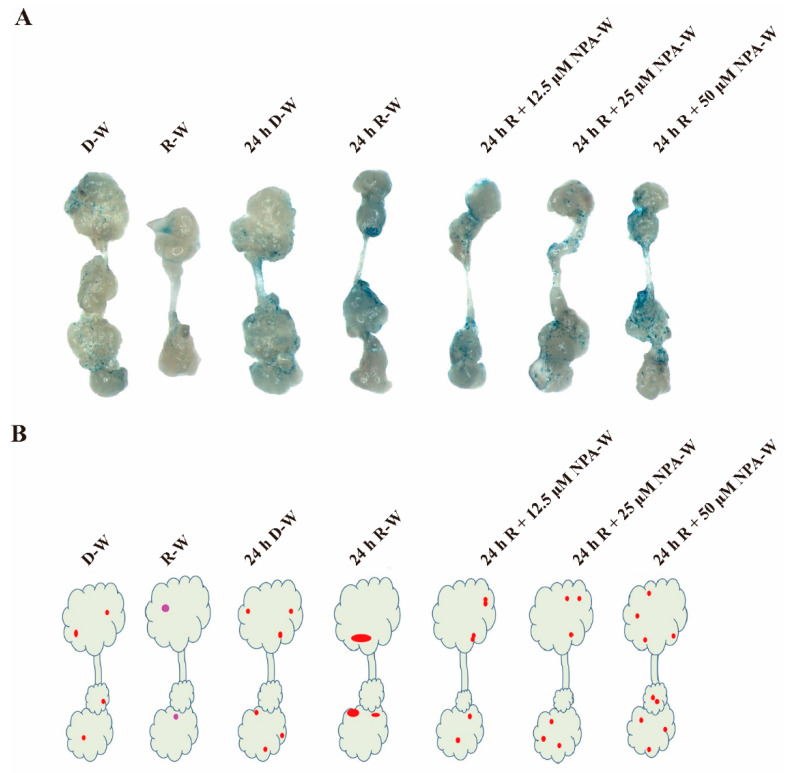
Dynamic localization and expression patterns of *WUS* under different light regimens and concentrations of NPA. (**A**) *WUS* localization patterns at day 7 on the SIM after different treatments. In contrast with D-W, the R-W treatment weakened the *WUS* signal, the 24 h R-W treatment promoted the centralized localization in the middle of the explant, the 24 R-NPA-W treatment increased the *WUS* signal and promoted a wider distribution of *WUS*. (**B**) A model for the *WUS* distribution pattern under different treatments. Red spots indicate strong *WUS* signal, pink spots indicate weak *WUS* signal. The size of spots indicates the area of *WUS* signal. Three biological replicates were performed for each experiment, and each replicate contained 120 root segments. Calluses derived from root segment were stained.

**Figure 5 plants-09-01378-f005:**
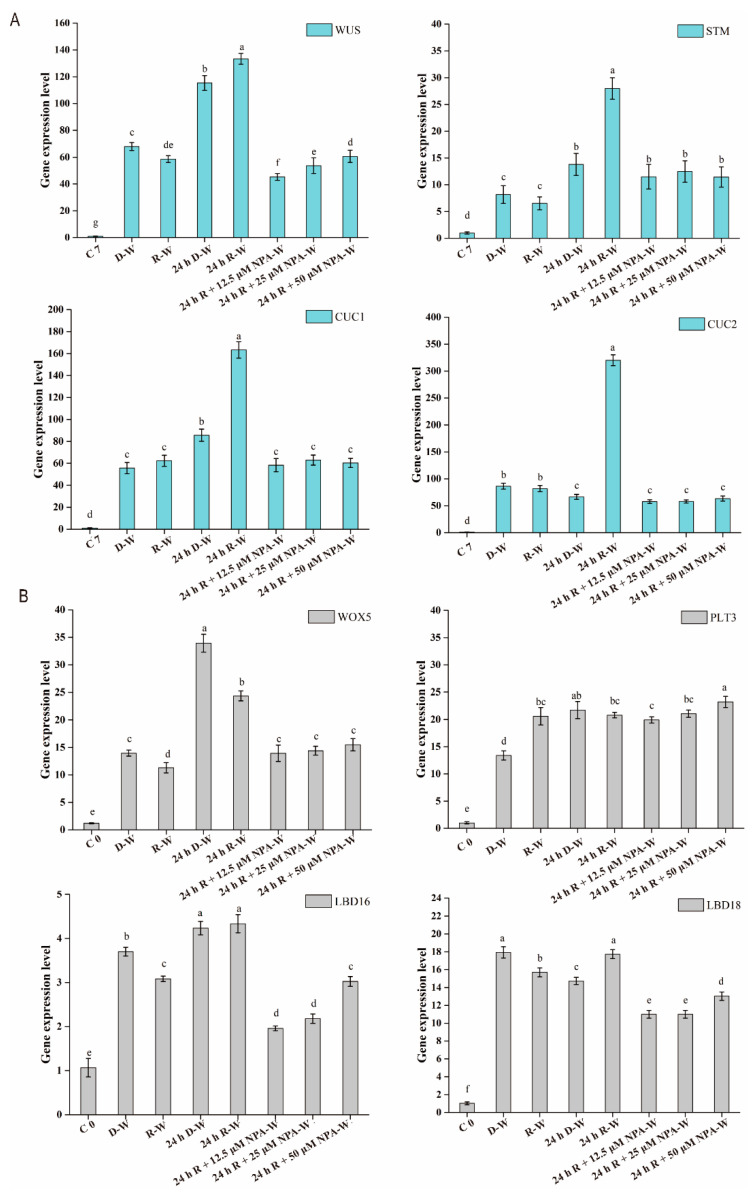
Expression patterns of shoot regeneration marker genes and callus development marker genes under different treatments. (**A**) Expression patterns of marker genes associated with shoot regeneration at day 7 on the CIM and SIM, respectively. (**B**) Expression patterns of marker genes involved in callus development at 7 day on the CIM. C0, root explants on the CIM at day 0, gene expression levels in excised roots at CIM day 0 was set to 1 for quantification of relative expression. C7, root explants on the CIM at day 7, gene expression levels in excised roots at CIM day 7 was set to 1 for quantification of relative expression. Error bars indicate the standard deviation from three independent experiments. A total of 30 individual calluses were collected for qPCR expression analysis for each biological replicate. The least significant difference method (LSD) was used for a significance test (*p* < 0.05); different lowercase letters represent statistical differences in pairwise comparisons between LSD test groups (*p* < 0.05).

**Figure 6 plants-09-01378-f006:**
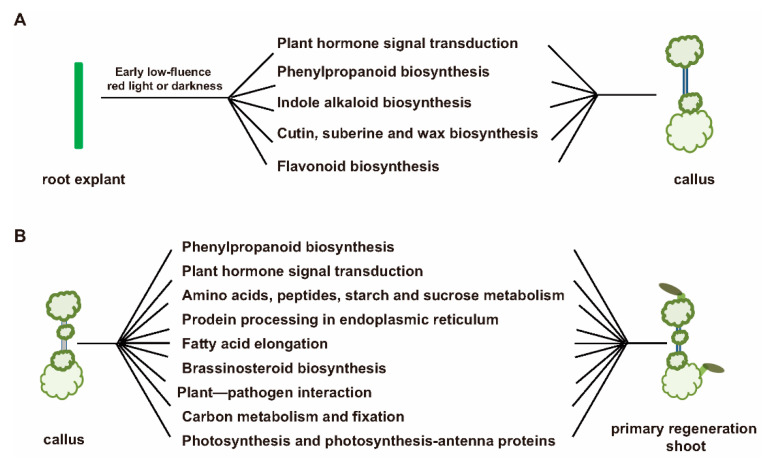
Functional enrichment of differential genes in the CIM and SIM stages under the early low-fluence red light or darkness. (**A**) Differential gene enrichment analysis revealed the biological pathway for the transition from root explant to callus in DCIM7 vs. RCIM7, DWCIM7 vs. DCIM7 and DWCIM7 vs. RCIM7. (**B**) Differential gene enrichment analysis revealed the biological pathway for the transition from callus to primary regeneration shoot in the SIM stages (DWSIM7 vs. DSIM7, DWSIM7 vs. RSIM7 and DSIM7 vs. RSIM7) and in the transitory stage from CIM to SIM (DCIM7 vs. DSIM7 and RCIM7 vs. RSIM7). RCIM7 (24 h R-W treatment, CIM 7 d); DCIM7 (24 h D-W treatment, CIM 7 d); DWCIM7 (D-W treatment, CIM 7 d); RSIM7 (24 h R-W treatment, SIM 7 d); DSIM7 (24 h D-W treatment, SIM 7 d) and DWSIM7 (D-W treatment, SIM 7 d); D-W (the control treatment); 24 h D-W, early 24 h dark and then shifting to 6 days white light in CIM followed by white light throughout SIM; 24 h R-W, early 24 h red light shifting to 6 days white light in CIM, followed by white light treatment in SIM; CIM, callus induction medium; SIM, shoot induction medium.

**Table 1 plants-09-01378-t001:** Significant representative differentially expressed genes (DEGs) involved in KEGG enrichment during the transitional stage (DCIM7 vs. DSIM7, RCIM7 vs. RSIM7).

Gene ID	Gene Name	Pathway	KO ID	Corrected *p*-Value	Rich Facter
AT1G02850	*BGLU11*	Phenylpropanoid biosynthesis	ko00940	6.15 × 10^−8^	1.96
AT1G15820	*CP24*	Photosynthesis—antenna proteins	ko00196	1.00 × 10^−5^	3.41
AT1G03130	*PSAD2*	Photosynthesis	ko00195	3.84 × 10^−5^	2.18
AT1G02850	*BGLU11*	Starch and sucrose metabolism	ko00500	4.14 × 10^−3^	1.53
AT1G01120	*KCS1*	Fatty acid elongation	ko00062	5.64 × 10^−3^	2.40
AT2G26710	*BAS1*	Brassinosteroid biosynthesis	ko00905	6.11 × 10^−3^	3.92
AT1G04240	*IAA3*	Plant hormone signal transduction	ko04075	1.31 × 10^−2^	1.42
AT1G12900	*GAPA2*	Carbon fixation in photosynthetic organisms	ko00710	1.52 × 10^−2^	1.89
AT1G09420	*G6PD4*	Carbon metabolism	ko01200	0.02	1.42
AT1G02850	*BGLU11*	Cyanoamino acid metabolism	ko00460	0.03	1.91
AT1G12550	*HPR3*	Glyoxylate and dicarboxylate metabolism	ko00630	0.03	1.80
AT2G19190	*FRK1*	Plant–pathogen interaction	ko04626	3.62	1.51

Corrected *p*-value ≤ 0.05; RCIM7 (24 h R-W treatment, CIM 7 d); DCIM7 (24 h D-W treatment, CIM 7 d); RSIM7 (24 h R-W treatment, SIM 7 d); DSIM7 (24 h D-W treatment, SIM 7 d); 24 h D-W, early 24 h dark and then shifting to 6 days white light in CIM followed by white light throughout SIM; 24 h R-W, early 24 h red light shifting to 6 days white light in CIM, followed by white light treatment in SIM; CIM, callus induction medium; SIM, shoot induction medium; KO, KEGG Ortholog.

**Table 2 plants-09-01378-t002:** Significant representative differentially expressed genes (DEGs) involved in KEGG enrichment during the dedifferentiation stage (DWCIM7 vs. DCIM7, DCIM7 vs. RCIM7, DWCIM7 vs. RCIM7).

Gene ID	Gene Name	Pathway	KO ID	Corrected *p*-Value	Rich Facter
AT2G40890	*REF8*	Flavonoid biosynthesis	ko00941	0.04	7.14
AT3G44540	*FAR4*	Cutin, suberine and wax biosynthesis	ko00073	4.66 × 10^−3^	6.78
AT1G74000	*SS3*	Indole alkaloid biosynthesis	ko00901	1.47 × 10^−3^	20.00
AT1G51680	*4CL1*	Phenylalanine metabolism	ko00360	6.07 × 10^−6^	7.86
AT1G05260	*RCI3*	Phenylpropanoid biosynthesis	ko00940	0	6.75

Corrected *p*-value ≤ 0.05; RCIM7 (24 h R-W treatment, CIM 7 d); DCIM7 (24 h D-W treatment, CIM 7 d); DWCIM7 (D-W treatment, CIM 7 d); D-W (the control treatment); 24 h D-W, early 24 h dark and then shifting to 6 days white light in CIM followed by white light throughout SIM; 24 h R-W, early 24 h red light shifting to 6 days white light in CIM, followed by white light treatment in SIM; CIM, callus induction medium; SIM, shoot induction medium; KO, KEGG Ortholog.

**Table 3 plants-09-01378-t003:** Significant representative differentially expressed genes (DEGs) involved in KEGG enrichment during the primary regeneration shoot stage (DWSIM7 vs. DSIM7, DWSIM7 vs. RSIM7, DSIM7 vs. RSIM7).

Gene ID	Gene Name	Pathway	KO ID	Corrected *p*-Value	Rich Facter
AT1G26560	*BGLU40*	Phenylpropanoid biosynthesis	ko00940	2.05 × 10^−5^	3.14
AT1G04980	*PDI10*	Protein processing in endoplasmic reticulum	ko04141	3.76 × 10^−3^	2.37
AT1G17420	*LOX3*	alpha-Linolenic acid metabolism	ko00592	5.59 × 10^−3^	4.89
AT1G06020	*FRK3*	Starch and sucrose metabolism	ko00500	1.94 × 10^−2^	2.23
AT1G72450	*JAZ6*	Plant hormone signal transduction	ko04075	0.10	1.89
AT4G28720	*YUC8*	Tryptophan metabolism	ko00380	0.16	3.49
AT1G02920	*GST11*	Glutathione metabolism	ko00480	0.19	2.59

Corrected *p*-value ≤ 0.05; RSIM7 (24 h R-W treatment, SIM 7 d); DSIM7 (24 h D-W treatment, SIM 7 d) and DWSIM7 (D-W treatment, SIM 7 d); D-W (the control treatment); 24 h D-W, early 24 h dark and then shifting to 6 days white light in CIM followed by white light throughout SIM; 24 h R-W, early 24 h red light shifting to 6 days white light in CIM, followed by white light treatment in SIM; CIM, callus induction medium; SIM, shoot induction medium; KO, KEGG Ortholog.

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
