# Peer review of "Early Low-Fluence Red Light or Darkness Modulates the Shoot Regeneration Capacity of Excised Arabidopsis Roots"

_plants, 2020, doi:10.3390/plants9101378_

Round 1

Reviewer 1 Report

It is an excellent work, very well structured and presented. I suggest to include a Conclusion section or paragraph to emphasize de discussion of the results. 

Graphics are really good and didactic. Please try to make a Graphical Abstract and include it because I am sure it will increase the impact of the publication.

Author Response

Dear Reviewer:

    First of all, Thank you for reviewing my manuscript. Our answer is in the attachment, please check the attachment.

Reviewer 2 Report

Review of the manuscript "Early low-fluence red light or darkness modulates the shoot regeneration capacity of excised Arabidopsis  roots".

The text submitted for review is a contribution to the study of applying differential light treatment for roots regenerative capacity excised from Arabidopsis thaliana wild-type Columbia (Col-0). The research material was properly selected, just like the experimental system was properly planned. Abstract was edited properly. Interesting and well presented results. The proper selection of literature. I only have some small remarks which I will point out below:

  1. Keywords should be separated by commas.
  2. The text of the manuscript, especially in terms of the Introduction, could be a little more precise (and refer to the experiments). The content on lines 30-45 may be significantly shortened.
  3. Just be careful when writing Latin names of species you had to use italics (such as Arabidopsis thaliana or as in line 411 Gossypium hirsutum).
  4. Please, correct the units from which the authors originate (in point 1 we write State Key Laboratory of Cotton Biology in capital letters).
  5. In materials and methods, I am not convinced by the PPFD unit notation used - please correct it (without the / sign).
  6. The records 80-90 (line 471) and 40-60 (line 474) are imprecise when it comes to treatment with white and red light, respectively. Please clarify this point.
  7. The section Conclusions is missing.

That's all the comments from me regarding this contribution.

Author Response

(The authors gave the same response as above.)
